# Fluorescent Aerolysin (FLAER) Binding Is Abnormally Low in the Clonal Precursors of Acute Leukemias, with Binding Particularly Low or Absent in Acute Promyelocytic Leukemia

**DOI:** 10.3390/ijms252211898

**Published:** 2024-11-05

**Authors:** María Beatriz Álvarez Flores, María Sopeña Corvinos, Raquel Guillén Santos, Fernando Cava Valenciano

**Affiliations:** 1Centro Nacional de Investigaciones Cardiovasculares Carlos III, 28029 Madrid, Spain; 2URSalud Laboratory, Hospital Universitario Infanta Sofia, 28702 San Sebastián de los Reyes, Spain; msopena@ursalud.com (M.S.C.); rguillen@salud.madrid.org (R.G.S.); ferjulte@gmail.com (F.C.V.)

**Keywords:** acute promyelocytic leukemia, FLAER, acute myeloid leukemia, flow cytometry, minimal measurable disease, PML-RARA, immunophenotype, paroxysmal nocturnal hemoglobinuria

## Abstract

Flow cytometry plays a fundamental role in the diagnosis of leukemias and lymphomas, as well as in the follow-up and evaluation of minimally measurable disease after treatment. In some instances, such as in the case of acute promyelocytic leukemia (APL), rapid diagnosis is required to avoid death due to serious blood clotting or bleeding complications. Given that promyelocytes do not express the glycophosphatidylinositol (GPI)-anchored protein CD16 and that deficient CD16 expression is a feature of some CD16 polymorphisms and paroxysmal nocturnal hemoglobinuria (PNH), we included the GPI anchor probe FLAER aerolysin in the APL flow cytometry probe panel. Initial tests showed that FLAER binding was absent in pathological promyelocytes from APL patients but was consistently detected with high intensity in healthy promyelocytes from control bone marrow. FLAER binding was studied in 71 hematologic malignancies. Appropriate control cells were obtained from 16 bone marrow samples from patients with idiopathic thrombocytopenic purpura and non-infiltrated non-Hodgkin’s lymphoma. Compared with the positive FLAER signal in promyelocytes from healthy bone marrow, malignant promyelocytes from APL patients showed weak or negative FLAER binding. The FLAER signal in APL promyelocytes was also lower than that in control myeloid progenitors and precursors from patients with other forms of acute myeloid leukemia (AML), B-cell acute lymphoblastic leukemia, or myelodysplastic syndrome. Minimal measurable disease studies performed in APL patients after treatment found normal promyelocyte expression when minimal measurable disease was negative and FLAER-negative promyelocytes when disease relapse was detected. The inclusion of FLAER in the flow cytometry diagnosis and follow-up of APL could be very helpful. Decreased FLAER binding was found in all cases of APL, confirmed by the detection of the PML-RARA fusion transcript and, to a lesser extent, in the other AMLs studied. This study also revealed FLAER differences in other acute leukemias and even between different precursors (myeloid and lymphoid) from healthy controls. However, the reason for FLAER’s non-binding to the malignant precursors of these leukemias remains unknown, and future studies should explore the possible relation with an immune escape phenomenon in these leukemias.

## 1. Introduction

The diagnosis of acute leukemia requires an integrated study that includes cell morphology, molecular and genetic analysis, and immunophenotyping. Flow cytometry (FCM) is useful not only for the diagnosis of acute leukemia but also for its classification and monitoring and for the evaluation of minimal measurable disease after treatment [1,2,3,4]. Technological advances in multiparametric FCM make it possible to achieve high sensitivity, and the speed of the technique allows results to be obtained rapidly, making flow cytometry an indispensable technique in the clinical laboratory. The speed of FCM analysis is particularly relevant to the diagnosis of acute leukemia, where correct identification of blast lineage is essential, especially for the diagnosis of acute leukemias that require rapid clinical assistance, such as acute promyelocytic leukemia (APL) [4,5].

APL is characterized by the production of a fusion transcript of promyelocytic leukemia (*PML*) and retinoic acid receptor alpha (*RARA*) genes and is now the most frequently curable acute leukemia in adults. However, successful treatment of APL requires prompt diagnosis and treatment initiation because patients can develop serious blood clotting or bleeding complications, and most deaths occur within the first month after diagnosis [6,7,8]. APL diagnosis has traditionally relied on the morphological identification of the leukemic cells, but for a rapid diagnosis, FCM is widely used because a multiparametric FCM immunophenotyping panel is generally sufficient to differentiate APL from other subtypes of acute myeloid leukemia (AML). Atypical promyelocytes have a highly specific immunophenotype: CD15−, CD16−, CD11b−, CD11c−, CD13+, CD33+, CD45+, CD64+/−, CD117+, and HLA-DR– [8,9], and a number of variant patterns have been described [5,8,9]. However, WHO criteria stipulate the need to confirm the FCM-based diagnosis by molecular cytogenetic detection of translocation of chromosomes 15 and 17 involving the *PML* and *RARA* genes (t(15;17)) or other rare variant translocations by karyotyping, fluorescence in situ hybridization (FISH), and reverse transcriptase–polymerase chain reaction (RT-PCR) [10,11,12].

The promyelocytes that accumulate in APL are arrested at an early stage of granulocyte development and lack expression of the GPI-anchored cell-surface antigen CD16 (Fcγ receptor III, FcγRIII). Absent or weak CD16 expression is also observed in myelodysplastic syndrome (MDS) and paroxysmal nocturnal hemoglobinuria (PNH), as well as in individuals genetically deficient for CD16 [13,14]. PNH is caused by a somatic mutation in the X-linked PIG-A gene, which encodes an enzyme essential for GPI synthesis [15,16]. The FCM-based diagnosis of PNH includes negative leukocyte staining with fluorescent aerolysin (FLAER), a fluorochrome-conjugated inactive variant of the bacterially derived channel-forming protein aerolysin, which specifically binds to GPI anchors [17,18,19,20].

In the analysis of a case of suspected acute leukemia involving the absence of CD16 expression on mature neutrophils in peripheral blood, we incorporated FLAER into our APL study panel to expedite results and simultaneously exclude the possibility of PNH. Based on the findings, we then decided to include FLAER analysis in the complete acute leukemia panel. Whereas promyelocytes from control patients were positive for FLAER binding, the malignant promyelocytes from APL patients were consistently negative. All FLAER-negative promyelocytes were positive for t(15;17), detected by FISH and/or PCR. Inclusion of FLAER in the follow-up of patients treated with all-trans retinoic acid (ATRA) revealed positive FLAER staining, coinciding in all cases with a negative t(15;17) test and thus confirming the absence of malignant promyelocytes. FLAER thus shows promise not only for APL diagnosis but also for the monitoring of minimally measurable disease after therapy. The inclusion of FLAER in the APL FCM study panel worked in bone marrow, peripheral blood, and cerebrospinal fluid samples. The diagnostic potential of FLAER binding was also analyzed in the malignant precursors of other acute leukemias.

## 2. Results

### 2.1. Comparison of FLAER Binding in Non-Malignant Promyelocytes and B-Cell Precursors from Control Patients Versus Malignant Promyelocytes and B-ALL Lymphoid Precursors

Having initially detected weak binding of FLAER to promyelocytes from APL patients, we compared the binding of FLAER to malignant APL promyelocytes with that to non-malignant precursors from control bone marrow samples. This analysis revealed a significant depletion of FLAER binding in APL promyelocytes (MFI = 1020.43) versus non-malignant promyelocytes (MFI = 20,011.00) and other myeloid precursors (MFI = 5707.33) (Mann–Whitney U *p* < 0.0001) (Figure 1, myeloid component). FLAER binding was also lower in B-cell acute lymphoblastic leukemia (B-ALL) precursors (MFI = 3894.43) than in CD34+ and CD34− B-lymphoid precursors from control bone marrow (MFI = 11,235.60 and 17,448.67, respectively; Mann–Whitney U *p* < 0.005) (Figure 1A, lymphoid component).

Analysis of FLAER binding to precursors from control bone marrow revealed a graduation from the lowest binding in myeloid precursors, followed by CD34+ B-cell precursors and CD34− B-cell precursors, and with the highest binding to normal promyelocytes (MFI = 5707.33, 11,235.60, 17,448.67, and 20,011.00, respectively; Mann–Whitney U *p* < 0.005).

Similar results were obtained with the FLAER ratio; however, this was less sensitive than the FLAER MFI for the detection of significant differences between B-ALL populations and CD34- B lymphoid precursors, as well as between control promyelocytes and CD34- B lymphoid precursors (Figure 1B).

All the control promyelocyte populations studied were positive for FLAER binding, contrasting with consistent non-binding to promyelocytes from APL patients (Figure 2).

### 2.2. FLAER Expression in Acute Leukemias

Given the results found in APL, we investigated FLAER binding to malignant precursors of other acute leukemias and the myeloid precursors of MDS. For ease of understanding and statistical analysis, cases were grouped according to the FAB classification (Table 1).

APL promyelocytes exhibited significantly lower FLAER binding (MFI) than observed in other AMLs and MDS (Figure 3A). FLAER binding differed significantly between FAB groups and with respect to control myeloid precursors. With the exception of AMLs with a monocytoid component (M4–M5), the AML clones studied had lower FLAER intensity than myeloid precursors from control bone marrow. Although FLAER binding was lower in FAB M0-M1 than in FAB M2, this difference was not statistically significant (*p* = 0.081), possibly reflecting the low numbers of M2 samples. The FLAER ratio showed similar results but did not reveal more significant statistical differences than FLAER MFI (Figure 3B).

### 2.3. Utility of FLAER Binding for the Study of Minimal Measurable Disease in APL

The management of APL has been revolutionized by treatment with ATRA, which triggers the differentiation of leukemic blasts into mature granulocytes, followed by spontaneous apoptosis of differentiated malignant cells [21]. A repeat analysis of promyelocytes in 7 patients who completed remission induction therapy with ATRA showed a complete restoration of the normal phenotype that was accompanied by positive FLAER staining (Figure 4). This phenotypic transition correlated in all cases with the absence of t(15;17), detected by FISH and/or PCR. This result indicates the utility of including FLAER in the APL FCM panel for the study of minimally measurable disease.

### 2.4. Relative Differences in FLAER Binding

Comparison of the FLAER signal in the different AML samples with that in myeloid precursors from control bone marrow showed that none of the AML samples bound FLAER with greater intensity than the control myeloid precursors. The weakest binding was found in APL (M3) samples (0.18 of the control signal), followed by M2 at 0.37, M0–M1 at 0.49, AML sec at 0.66, MDS at 0.88, and M4–M5 at 0.93 (Figure 5).

### 2.5. FLAER MFI Algorithm for the Study of APL

Although APL had the lowest FLAER binding of all AML types studied, this signal did not differ significantly from that of M0-M1 samples. Unfortunately, the FCM diagnosis of leukemias and lymphomas is not achieved with a single diagnostic marker, and this case is no exception. Nevertheless, based on our findings, we designed the following algorithm that incorporates FLAER binding to identify cases of APL with greater accuracy (Figure 6).

In the proposed workflow, samples from patients with suspected APL are screened with an FCM panel that includes FLAER. If the immunophenotyping of the pathological cells suggests APL, the FLAER MFI is assessed. Cells that do not present an APL AML phenotype or have a FLAER MFI > 2500 are excluded from the suspicion of APL. Pathological cells with a FLAER MFI ≤ 2500 are further analyzed for the CD13/CD33 expression (see Figure 7A). A compatible CD13/CD33 pattern would strongly suggest a probable diagnosis of APL. Otherwise, APL would be considered unlikely, although a PML-RARA rearrangement analysis would still be recommended to ensure that no case of APL-derived AML is excluded.

Of the 70 suspected acute leukemia analyzed here, patient samples from 19 contained a precursor population with an APL-like phenotype. Of these, 18 had a FLAER, MFI ≤ 2500. Of these 18 cases, 14 showed high expression of CD33 and, to a lesser extent, CD13, consistent with the previously described pattern, and were ultimately diagnosed as APL upon confirmation of the PML-RARA rearrangement. Of the remaining 4 cases, 3 were ultimately diagnosed as AML with minimal differentiation (M0 FAB group), and the other was diagnosed as AML without maturation (M1 FAB group) (Appendix A).

These results achieve a sensitivity and specificity of 100% in the diagnosis of APL. However, it is important to emphasize the need to confirm the consistency of these findings in a larger number of cases.

## 3. Discussion

PNH is caused by somatic mutations in the phosphatidylinositol glycan anchor biosynthesis class A (PIGA) gene. As a result, cells cannot express glycosylphosphatidylinositol-anchored proteins (GPI-APs) and thus lack the GPI-dependent cell membrane-bound antigens CD16, CD14, CD55, and CD59, among others. FLAER specifically binds to GPI-anchors and is extensively used in the FCM diagnosis of PNH, with the lack of a FLAER signal detecting a PNH clone. In the present study, we detected similarly low FLAER binding in clonal precursors of acute myeloid and lymphoid leukemias, with binding especially low in the malignant promyelocytes that accumulate in APL. These results are consistent with those described by Li, L et al. [22]. Analysis of myeloid and lymphoid precursors from control bone marrow showed that as these precursors mature, FLAER expression increases and that all stages of granulocyte maturation from promyelocyte to mature neutrophil bind FLAER with high intensity.

Of the acute leukemias studied, the only one in which the malignant cells showed high FLAER intensity was BPDCN; however, only 3 cases of this rare hemopathy were included in the study.

FCM-based diagnosis of leukemias and lymphomas requires more than one marker, and FLAER binding is thus insufficient by itself. However, the incorporation of FLAER MFI analysis in a workflow that also considers the CD13/CD33 pattern successfully classified all 14 cases of APL, with diagnosis ultimately confirmed by the detection of the PML-RARA rearrangement. The combined FLAER MFI and CD13/CD33 analysis thus provided 100% sensitivity and specificity. Although the robustness of the workflow algorithm will require confirmation in a larger sample number, the consistently negative FLAER signal in APL promyelocytes is a highly promising marker for the early diagnosis of APL and the detection of minimally measurable disease after treatment.

A further advantage is that FLAER MFI analysis detected larger differences between AML types than the FLAER ratio, simplifying the analysis and algorithm implementation by eliminating the need for calculations. Although we describe the cytometer calibration procedure to support the reproducibility of the results, we recommend that each laboratory establish its own FLAER MFI cutoff, as different cytometers with varying configurations may be in use.

Future research should investigate the underlying mechanisms of FLAER negativity in pathological precursors of acute leukemias, including the role of modulated GPI anchors and potential immune escape phenomena, which could suggest novel therapeutic targets. This study offers a detailed analysis of FLAER binding across various acute leukemias, including B-ALL, and extends to myelodysplastic syndromes (MDS).

It also provides a comprehensive assessment of FLAER binding in both myeloid and lymphoid progenitors in healthy controls. These results establish the value of FLAER MFI in the diagnosis of APL and highlight the need for further research into FLAER binding variations between distinct progenitor populations in a larger number of cases, with the aim of increasing the accuracy of FCM-based diagnostic and monitoring methodologies for acute leukemias.

## 4. Materials and Methods

### 4.1. Patients and Samples

Patient samples included in the diagnostic study were referred for immunophenotyping due to suspicion of acute leukemia. The inclusion criteria comprised a newly suspected diagnosis of acute leukemia and the availability of sufficient samples for FLAER analysis. Samples from patients with MDS were included to enhance the comprehensiveness of the analysis. No leukemia patients were excluded a priori. For the assessment of minimal measurable disease, FLAER analysis was conducted exclusively in samples from patients with acute promyelocytic leukemia (APL). Patient and laboratory characteristics related to the control and hematological malignancy samples included in the study are shown in Appendix A.

This study included 71 samples from patients with hematologic malignancies: 53 cases of acute myeloid leukemia (of which 14 were APL), 7 cases of acute B-lymphoblastic leukemia, 3 cases of blastic plasmacytoid dendritic cell neoplasm, and cases 8 of myelodysplastic neoplasm. Diagnosis and immunological classification were in accordance with WHO criteria, and diagnosis of APL was confirmed by detection of t(15;17). Fleshly obtained peripheral blood, EDTA-anticoagulated bone marrow, and non-anticoagulated cerebrospinal fluid were collected and examined within 12 h. The FCM probe panel included FLAER and antibodies to CD34, CD117, HLA-DR, CD11b, CD13, CD16, CD33, CD56, and CD64, plus additional antibodies for the characterization of specific leukemias and MDS (BD Biosciences, San Jose, CA; FLAER was kindly provided by Alexion and purchased from Protox Biotech, Victoria, BC, Canada). The most frequently used panel configuration is described in Appendix A. Control samples were prepared from bone marrow aspirates obtained from 12 patients with idiopathic thrombocytopenic purpura (ITP) and 4 patients with non-Hodgkin’s lymphoma with no bone marrow involvement.

### 4.2. Surface and Intracellular Staining

Bone marrow cells were dispensed and washed twice with phosphate-buffered saline (PBS) (500× *g* for 5 min). Supernatants were removed, and cells were incubated with appropriately tittered fluorescently labeled antibodies and FLAER (diluted 1/10 in PBS) for 15 min at room temperature (RT) in the dark. A total of 2 mL of lysis buffer was added to each sample (BD FACS™ Lysing Solution, BD Biosciences, San Jose, CA, USA), diluted 1:10 in distilled water. Samples were then vortexed gently, incubated for 6 min at RT, and centrifuged at 500× *g* for 5 min. The supernatant was removed, lysis was stopped by the addition of 4 mL PBS, and samples were centrifuged again and the supernatant removed. Finally, 0.5 mL of PBS was added, and the samples were analyzed by FCM, as described below.

For the intracellular staining of MPO, TdT, CD3, and other antigens as required, appropriately labeled antibodies were added after the lysis centrifugation, and the reaction was incubated for 10 min at RT, followed by an additional wash in PBS [23]. Because the lysis solution (BD FACS™ Lysing Solution, BD Biosciences, San Jose, CA, USA) contains formaldehyde and methanol, and these components confer permeabilizing properties, it was not necessary to use an additional permeabilizing solution.

### 4.3. Flow Cytometry

Eight-color FCM analysis was performed with a BD FACSCanto II™ flow cytometer (BD Biosciences, San Jose, CA, USA), and data were analyzed with Infinicyt™ software Version 2.0.6.b (BD Biosciences, San Jose, CA, USA, https://www.bdbiosciences.com/en-eu/products/software/infinicyt-software, last accessed at 16 of September 2024). In all cases, debris was removed by gating in an FSC/SSC dot plot, and doublets were discriminated in an FSC-Area/FSC-High dot plot.

### 4.4. Cytometer Standardization

Comparability of mean fluorescence intensity (MFI) data between patient samples over time was achieved using the BD Cytometer Setup and Tracking System (CS&T) and the BD FACSDiva™ Version 6 software recommendations.

Photomultiplier (PMT) voltages were adjusted to unlabeled lysed whole blood cells to obtain optimal PMT voltages for the resolution of dim cell populations. The target values resulting from the PMT optimization were used for subsequent calibrations to maintain instrument standardization [24]. Instrument configuration and mean fluorescence intensity target values are presented in Appendix A.

### 4.5. Determination of FLAER Binding in Healthy Promyelocytes, Malignant Promyelocytes, and Precursors from Other Acute Leukemias

Assessment of mean and median FLAER binding showed no notable differences, and median values were therefore used for analysis. To enable a comparison of FLAER binding between normal and pathological precursor populations, we implemented a calibration standardization utilizing the MFI targets detailed in Appendix A. This methodology ensures that observed differences in the signal are attributable to the populations themselves and not to the influence of varying equipment calibration conditions.

The MFI of bound FLAER was measured in healthy promyelocytes from 16 control samples and malignant promyelocytes from bone marrow aspirates of 14 APL patients. Promyelocytes were identified by the CD45+, CD16−, CD11b−, CD13+, CD33+, and HLA-DR-phenotypes and characteristic features of forward and side scatter distribution (Appendix A).

Precursors obtained from patients with other acute leukemias were identified according to the expression of CD117, CD34, CD45, and HLA-DR, and FLAER binding was then analyzed in the blast population and in appropriate precursor populations from control patients (Appendix A).

### 4.6. Assessment of the CD13/CD33 Expression Pattern

The pathological population was selected, and its expression profile was analyzed in comparison with a reference image overlay of myeloid precursors derived from control bone marrow. The expression pattern indicative of APL AML is characterized by higher intensity for CD33 and, to a lesser extent, CD13. In contrast, non-APL AML samples are characterized by a CD13/CD33 expression profile that does not fulfill these criteria and is thus incompatible with APL (Figure 7).

### 4.7. Determination of the FLAER Ratio

For the calculation of the FLAER ratio, the target population was selected, and the median fluorescence intensity (MFI) was analyzed. Simultaneously, the lymphoid population in the same sample was identified, and the MFI for this population was also noted. The FLAER ratio was calculated by dividing the FLAER MFI of the target population by the MFI of the corresponding lymphocytes. The ratio was calculated and compared in all controls and all AML cases; representative results are shown in Figure 8.

### 4.8. Statistical Analysis

The statistical significance of differences was determined with the Mann–Whitney U test or the two-tailed Student *t*-test, depending on the variable distribution. Statistical differences were considered significant at *p* < 0.05. * *p* < 0.05, ** *p* < 0.01, *** *p* < 0.001, and **** *p* < 0.0001. Statistical comparisons were performed with SPSS version 21.0 (IBM, Armonk, NY, USA), and some graphical representations were generated with GraphPad Prism version 10.2.3 (GraphPad Software, San Diego, CA, USA).

## Figures and Tables

**Figure 1 ijms-25-11898-f001:**
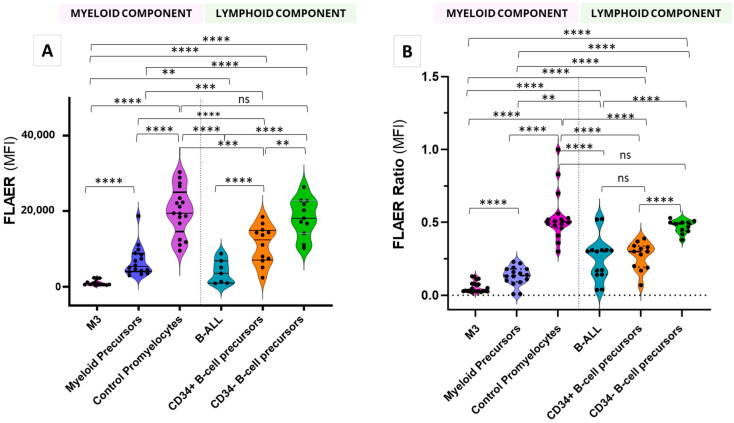
Differential FLAER means fluorescence intensity (MFI) (**A**) and FLAER ratio (**B**) in malignant promyelocytes from patients with acute promyelocytic leukemia (APL, M3), B-ALL, and control samples derived from bone marrow, including myeloid precursors, promyelocytes, and B-cell precursors. **** *p* < 0.0001; *** *p* < 0.005; ** *p* < 0.005; ns, non-significant (Mann Whitney U test). B-ALL, B-cell acute lymphoblastic leukemia; M3, APL with PML-RARA fusion (FAB classification).

**Figure 2 ijms-25-11898-f002:**
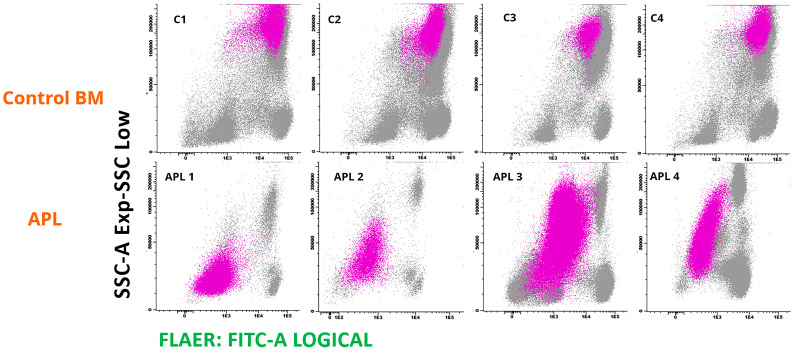
FLAER binding to promyelocytes from four control bone marrow samples (C1–C4, top row), contrasting with low binding to malignant promyelocytes from four bone marrow aspirates obtained from APL patients at diagnosis (APL 1–APL 4). Promyelocytes are depicted in fucsia.

**Figure 3 ijms-25-11898-f003:**
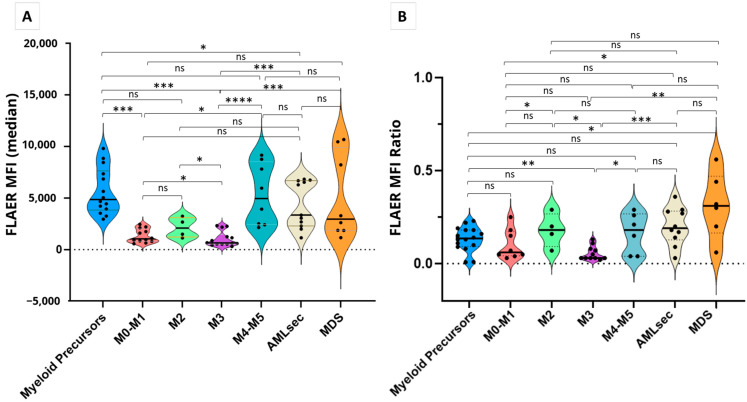
Comparison of FLAER MFI (**A**) and FLAER ratio (**B**) in the studied AMLs and control myeloid precursors. Control myeloid precursor samples were derived from bone marrow controls.**** *p* < 0.0001; *** *p* < 0.005; ** *p* < 0.005; * *p* < 0.05; ns, non-significant (Mann–Whitney U test).

**Figure 4 ijms-25-11898-f004:**
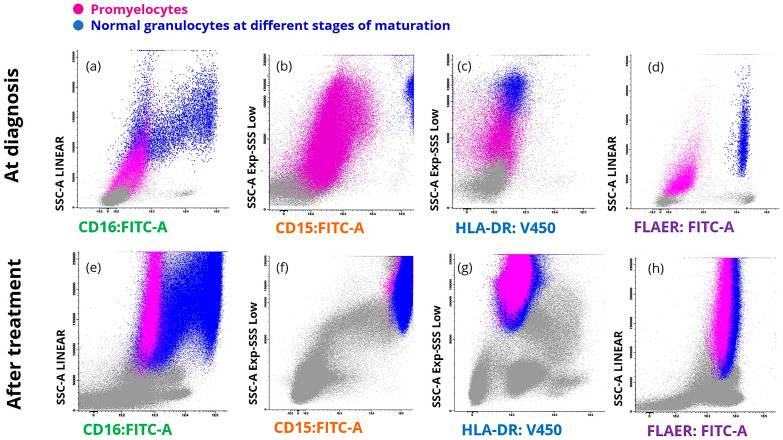
Representative promyelocyte analysis in a patient at the time of APL diagnosis (top row, (**a**–**d**)) and upon completion of remission induction therapy with ATRA (bottom row, (**e**,**f**)). The analysis shows cellular complexity in relation to CD16, CD15, and HLA-DR expression and FLAER binding. Recovery of cellular complexity after treatment is evident from the granulocytic component populations (**e**) and the increases in CD15 expression and FLAER binding (**f**,**h**). The HLA-DR expression remains negative, like normal promyelocytes (**g**).

**Figure 5 ijms-25-11898-f005:**
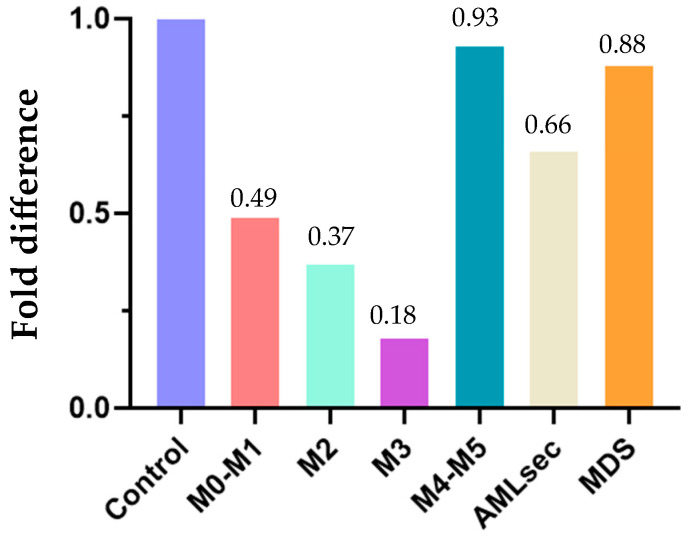
Differences in FLAER binding between AML samples relative to control myeloid precursors.

**Figure 6 ijms-25-11898-f006:**
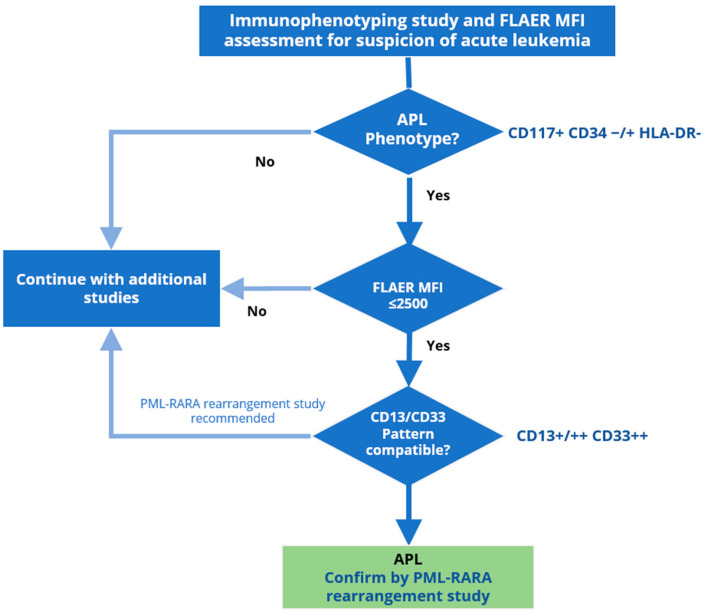
Proposed workflow algorithm for the identification of APL. If immunophenotyping of a suspected case of APL-derived AML identifies pathological cells with an APL-like phenotype, the FLAER MFI is assessed. Cells lacking an APL-like phenotype or having a FLAER MFI > 2500 exclude suspicion of APL. Conversely, if the FLAER MFI is ≤2500, the CD13/CD33 expression pattern is analyzed. A compatible expression pattern strongly suggests a diagnosis of APL, to be confirmed by PML-RARA rearrangement analysis. An incompatible CD13/CD33 pattern suggests that APL is unlikely; nevertheless, PML-RARA rearrangement analysis is recommended to exclude APL.

**Figure 7 ijms-25-11898-f007:**
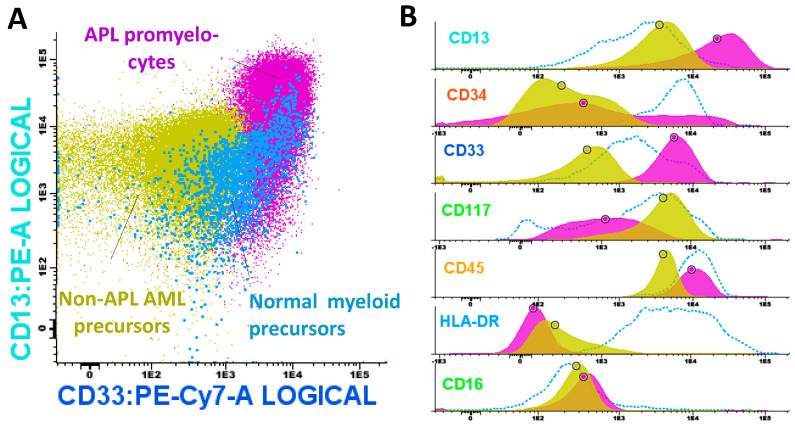
(**A**) Representative differential expression of CD13 and CD33 in normal myeloid precursors (cyan), promyelocytes from a patient with APL (fuchsia), and myeloid precursors with low FLAER MFI from a patient with non-promyelocytic AML (khaki). The pattern compatible with APL AML is characterized by high expression of CD33 and, to a lesser degree, CD13. (**B**) Histograms showing the expression of the APL phenotype (CD117+, CD34−/+, and HLA-DR-) alongside CD13, CD33, CD45, and CD16 in the defined populations. Circles denote median values.

**Figure 8 ijms-25-11898-f008:**
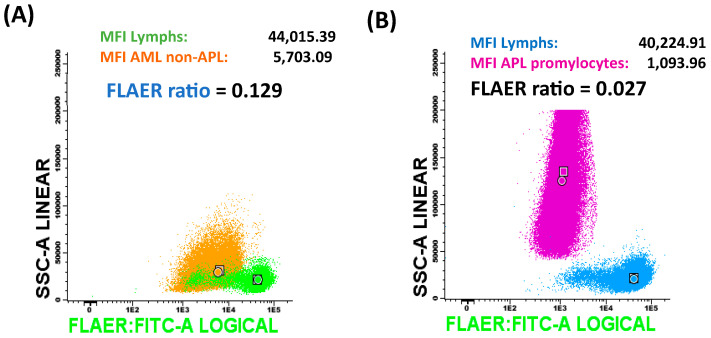
Representative SSC-A/FLAER plots in A) non-APL AML and B) APL AML. (**A**) The orange population corresponds to non-APL AML cells, while the green represents lymphocytes in the same sample. (**B**) Fuchsia denotes the APL promyelocyte population, and blue represents lymphocytes in the same sample. Circles correspond to median values, and the squares represent the mean fluorescence intensity of FLAER, confirming that they are comparable.

**Table 1 ijms-25-11898-t001:** Hematological neoplasm clustering and FLAER binding.

Neoplasms and Acute Leukemias (WHO Classification):	FAB Group	Mean FLAER MFI	N	Standard Deviation
AML with minimal differentiation and AML without maturation	M0–M1	2807.54	15	3255.01
AML with maturation	M2	2135.62	4	989.29
Acute promyelocytic leukemia with *PML-RARA* fusion	M3	1020.42	14	719.58
Acute myelomonocytic leukemia and Acute monocytic leukemia	M5-M5	5339.74	8	2967,06
AML, myelodysplasia-related	LMAsec	3776.92	11	2357,19
B-lymphoblastic leukemia	B-ALL	3894.51	7	3134,99
Blastic plasmacytoid dendritic cell neoplasm	DC Leuk	12,919.63	3	9714,28
Myelodysplastic neoplasm	MDS	5022.11	8	4067.57
Acute megakaryoblastic leukemia	M7	314.68	1	N/A
Controls				
Myeloid precursors	NA	5707.33	16	2403.04
CD34+ B precursors	NA	11,235.6	12	5086.74
CD34−B precursors	NA	17,448.67	12	6975.67
Healthy Promyelocytes	NA	20,011.00	16	6311.00

NA: not applicable.

## Data Availability

All data related to this study are presented in the article; further inquiries can be directed to the corresponding author.

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
