# Peer review of "Fluorescent Aerolysin (FLAER) Binding Is Abnormally Low in the Clonal Precursors of Acute Leukemias, with Binding Particularly Low or Absent in Acute Promyelocytic Leukemia"

_ijms, 2024, doi:10.3390/ijms252211898_

Round 1
Reviewer 1 Report
Comments and Suggestions for Authors
This is a very interesting and well constructed and presented study about the use of FLAER in the diagnosis and minimal measurable disease in acute leukemias and especially in APL.
The authors have created a well designed study with appropriate controls that prove the value of FLAER use in acute leukemias.
A clarification of the cytoplasmic staining method: don't the authors use any permeabilization reagent (l. 218-220)?
Have the authors compared FLAER binding in APLs with that in other AMLs with similar phenotype HLADR-, CD15- that are not proved as APLs by molecular techniques? If done, please, clarify it in the manuscript.
Comments on the Quality of English LanguageA few spelling mistakes should be corrected. A careful check of the manuscript is needed although English seems quite good and understandble.
e.g. l/14 case of APL (not with)
l. 212-213 the verb at the end of the sentence.
Author Response
Comment 1: This is a very interesting and well-constructed and presented study about the use of FLAER in the diagnosis and minimal measurable disease in acute leukemias and especially in APL.
The authors have created a well designed study with appropriate controls that prove the value of FLAER use in acute leukemias.
Response 1: We thank the reviewer for their positive evaluation of our manuscript.
Comment 2: A clarification of the cytoplasmic staining method: don't the authors use any permeabilization reagent (l. 218-220)?
Response 2: Yes, we use FACSLysing lysis solution, which, according to the manufacturer's specifications, contains 9.77% formaldehyde and 3.43% methanol. These components have permeabilizing properties, allowing us to label even nuclear antigens such as TdT. Below is an example showing positive MPO staining in pathological APL promyelocytes (fuchsia) compared to negative staining in the lymphoid population (green). Please see attached file.
Comment 3: Have the authors compared FLAER binding in APLs with that in other AMLs with similar phenotype HLADR-, CD15- that are not proved as APLs by molecular techniques? If done, please, clarify it in the manuscript.
Response 3: We thank the reviewer for this insightful comment, which has prompted us to conduct a more in-depth analysis of these cases. Through the application of a novel workflow algorithm (Figure 6) incorporating the CD13/CD33 expression pattern, we have been able to discriminate with 100% sensitivity and specificity between AML cases showing an APL-like phenotype but ultimately not diagnosed as APL, and cases confirmed to be APL. These findings were corroborated through detection of the PML-RARA rearrangement. This information has been incorporated into the manuscript; however, we acknowledge that the inclusion of additional cases would help to validate the robustness of the algorithm. Please, see figure 6 in the attached file.
Comments on the Quality of English Language
Comment 4: A few spelling mistakes should be corrected. A careful check of the manuscript is needed although English seems quite good and understandble.
Response 4: e.g. l/14 case of APL (not with) Corrected.
- 212-213 the verb at the end of the sentence. I am very sorry, I have not been able to locate this error.

Reviewer 2 Report
Comments and Suggestions for Authors
In the study by María Beatriz Álvarez Flores and co-authors, FLEAR was tested as a surrogate marker of APL. As flow cytometry is a method of choice for haemato-oncology diagnosis of leukaemia, improvements in commonly used antibody panels seem promising for increasing accuracy and precision. Thus, the scientific sound of the presented study is highly valid. The paper has an excellent introduction and gives essential information on FLEAR and reasons for testing it in leukaemias. Unfortunately, some flaws limit the significance of the study.
1. No clinical and demographic characteristics of participants were given. Please refer to the study's inclusion/exclusion criteria.
2. The FLEAR results seemed to be given as a row of MFI values. An index based on an unstained/negative population would be better for clarity.
3. As the study shows FLEAR to be an APL diagnosis marker, ROC analysis, sensitivity, specificity, and suggested cut-off should be given.
4. For CSF sample testing, only one replicate was used. Thus, please consider omitting this aspect of the study. The results are still promising even despite no CSF assay characteristics.
5. Please consider to update the the title as “preliminary report”
Author Response
Comment 1: In the study by María Beatriz Álvarez Flores and co-authors, FLEAR was tested as a surrogate marker of APL. As flow cytometry is a method of choice for haemato-oncology diagnosis of leukaemia, improvements in commonly used antibody panels seem promising for increasing accuracy and precision. Thus, the scientific sound of the presented study is highly valid. The paper has an excellent introduction and gives essential information on FLEAR and reasons for testing it in leukaemias.
Response 1: Thank you very much for this comment.
Comment 2: Unfortunately, some flaws limit the significance of the study. No clinical and demographic characteristics of participants were given. Please refer to the study's inclusion/exclusion criteria.
Response 2: You are correct; the following paragraph has been included in the Patients and Samples section, and a supplementary table has been added containing data on age, sex, leukocyte count, hemoglobin levels, and platelet counts.
“Patient samples included in the diagnostic study were referred for immunophenotyping due to suspicion of acute leukemia. The inclusion criteria comprised a newly suspected diagnosis of acute leukemia and the availability of sufficient sample for FLAER analysis. Samples from patients with MDS were included to enhance the comprehensiveness of the analysis. No leukemia patients were excluded a priori. For the assessment of minimal measurable disease, FLAER analysis was conducted exclusively in samples from patients with acute promyelocytic leukemia (APL). Patient and laboratory characteristics related to the control and hematological malignancy samples included in the study are shown in Table S1.” Please, see Table S1 in the attached file.
Comment 3: The FLEAR results seemed to be given as a row of MFI values. An index based on an unstained/negative population would be better for clarity.
Response 3: Thank you very much for your observation. Assessment of mean and median FLAER binding showed no notable differences, and median values were therefore used for analysis. To enable comparison of FLAER binding between normal and pathological precursor populations, we implemented a calibration standardization utilizing the MFI targets detailed in new Table S3 (below). This methodology ensures that observed differences in the signal are attributable to the populations themselves, and not to the influence of varying equipment calibration conditions. See table S3 in the attached file.
We now represent both the FLAER MFI for the acute leukemia precursor populations and also the ratio of these values to FLAER binding in the lymphocyte population in the same sample (figures 1 and 3).
Although differences were noted, the ratio did not improve the discrimination among the studied groups. The procedure for calculating the ratio is detailed in the Materials and Methods section (Determination of the FLAER Ratio) and a representative calculation is shown in Figure 7.
Additionally, we conducted a study of relative FLAER binding to clarify the differences between groups, again demonstrating that, of all the AML types studied, APL AML has lowest FLAER binding relative to the control group (normal myeloid precursors). This comparison is described in Result section 2.4 “Relative differences in FLAER binding”.
Comment 4: As the study shows FLEAR to be an APL diagnosis marker, ROC analysis, sensitivity, specificity, and suggested cut-off should be given. In light of your comments and those of Reviewer 1, we have expanded the study by developing a workflow algorithm to discriminate between APL and cases of AML with an APL-like phenotype based on FLAER MFI and the CD13/CD33 expression pattern (fig 6). This workflow algorithm enabled the identification of APL AML cases with 100% sensitivity and specificity, and the results were confirmed by detection of the PML-RARA rearrangement. Although the sample collection included only four cases showing an APL-like phenotype and low FLAER MFI, but an incompatible CD13/CD33 pattern, the contribution of this algorithm to identifying APL cases is noteworthy. The proposed workflow algorithm is detailed in section 2.5, “FLAER MFI Algorithm for the Study of APL”, and the analysis of the CD13/CD33 pattern is described in the Materials and Methods section. The analysis of the CD13/CD33 pattern in four representative APL cases and the four cases with an APL-like phenotype and low FLAER but an incompatible CD13/CD33 pattern is presented Figure S1.
Comment 5: For CSF sample testing, only one replicate was used. Thus, please consider omitting this aspect of the study. The results are still promising even despite no CSF assay characteristics.
Response 5: This information has been omitted.
Comment 6: Please consider to update the the title as “preliminary report”.
Response 6: Considering the new information provided, the manuscript now describes the procedures more comprehensively, and the new results lend greater robustness to the study. Therefore, it may no longer be necessary to include “preliminary report” in the title.

Round 2
Reviewer 2 Report
Comments and Suggestions for Authors
The authors have improved the text accordingly. I have no more questions.